# Understanding risk factors for disordered eating symptomatology in athletes: A prospective study

**Hannah Stoyel**[1]*, **Chris Stride**[2], **Vaithehy Shanmuganathan-Felton**[3], **Lucy Serpell**[1]

**1** University College London, London, England, United Kingdom, **2** University of Sheffield, Sheffield, England, United Kingdom, **3** University of Roehampton, London, England, United Kingdom

* hannah.stoyel.13@ucl.ac.uk

**Data Availability Statement:** All relevant data are within the manuscript and its Supporting Information files.

## Abstract

Disordered eating and eating disorders have huge impact on athletic health and performance. Understanding risk factors for disordered eating development is paramount to protecting the health and performance of these athletes. This project tested a model longitudinally to test whether body dissatisfaction (mediated by negative affect) and societal pressures (mediated by internalisation) predicted bulimic symptomatology at 1 year. The study recruited 1017 male and female athletes in a range of sports at three time points over a year. Cross-lag meditation modelling in MPLUS was utilised to test the hypothesised model. Results indicated that societal pressures mediated by general internalisation led to bulimic symptomatology and that gender and sport type do moderate the relationships. However, measurement issues indicate that scales not originally created for athletes may not reliably measure athletes' experience. This research highlights how understanding how to better assess risk factors and disordered eating related concepts in athletes is a key next step. The study is unique in its longitudinal design and in its sampling of a wide range of sports in both male and female athletes.

## Introduction

Eating disorders, including Anorexia Nervosa, Bulimia Nervosa and Binge Eating Disorder, are psychiatric illnesses that have significant negative impact on physical and mental well-being [1–3]. Eating disorders sit at the end of a continuum that also includes healthy or intuitive eating at the other end, and disordered eating in the middle [4, 5]. Disordered eating, defined as a subclinical level of issues with food restriction, bingeing and purging behaviours [6] has a much higher prevalence rate than clinically diagnosable eating disorders in both the general and sporting population [7–9].

Exercise is often a symptom or maintenance factor for an eating disorder, however, it is also an integral part of sport [10, 11], which means it is difficult to distinguish pathological from 'normal' exercise in athletes. Furthermore, the link between body shape and physical exercise is undeniable, making sport a complex feeding ground for disordered eating

**Funding:** No funding was provided for the study from any grants or organizations.

**Competing interests:** The authors have declared that no competing interests exist.

development [12–14]. Biomechanical differences between sports demand specific body shapes to increase the chance for successful performance, such as broad shoulders for rowing and minimal body mass for distance running. As such, it is often expected that athletes will have a higher prevalence rate of disordered eating than nonathletes, though rates appear to vary dependent on gender and sport type [15–18]. Athletes who participants in lean sports, those in which success depends on a lean body shape, are at a greater risk [6, 19–25]. Gender plays a moderating role, with prevalence higher for males in anti-gravitational sports such as cycling, and for females in lean or aesthetic sports such as dance [12, 26, 27].

Eating disorders and disordered eating can have long term impact on both health and on quality of life, and as such, understanding how to predict the development of eating disorders and disordered eating to facilitate prevention is paramount [9, 28, 29]. In 2007, Petrie and Greenleaf—based on Stice's [1994] dual pathway model of predicting bulimia—developed a detailed model that aims to predict disordered eating in athletes [30]. In Petrie and Greenleaf's model there are eight risk factors discussed: Sport pressures and societal pressures are the predictors, internalisation, body dissatisfaction, negative affect, modelled behaviours, which are thought to mediate the pathway to disordered eating symptomatology, which is deconstructed into restrained eating and binge eating and bulimia.

Stoyel and colleagues conducted a systematic review of the literature testing tenets of Petrie and Greenleaf's model, which indicated broad support for many of the relationships posited in the model [18]. However, contradictory findings between studies were due to the range of different designs, samples and measurement tools. Only three were of longitudinal design meaning that it was difficult to investigate causation. In general, there has been a lack of longitudinal research in this topic area.

Based on this review, Stoyel and colleagues tested Petrie and Greenleaf's model using structural equation modelling on a large dataset of male and female athletes from multiple sports [31]. Petrie and Greenleaf's model offered an inadequate fit to the data: using theoretically-driven adjustments, a new, more parsimonious model was developed [Fig 1], which offered a satisfactory fit. This revised model, hereafter known as the T1 model, is similar to Stice's 1994 dual pathway model; body dissatisfaction and social pressures predict bulimic symptomatology in athletes, with these effects operating indirectly via the twin mediators of negative affect and internalisation respectively [31].

While the T1 model provided some insight into how disordered eating may develop in athletes, it employed a cross-sectional design. Therefore, the aim of the current study was to test the utility and applicability of the T1 model employing a longitudinal design–by collecting

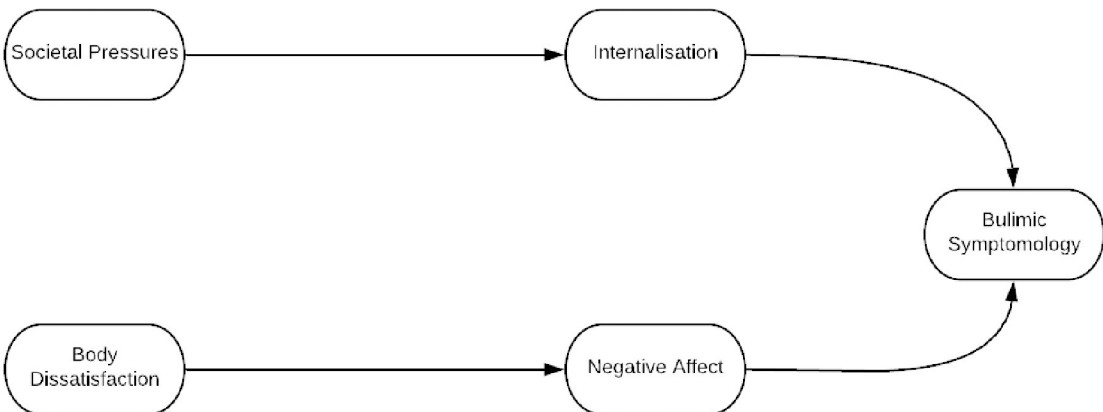

**Fig 1. T1 model.** This model has been adapted slightly from its original published version. Social pressures is now termed Societal Pressures.

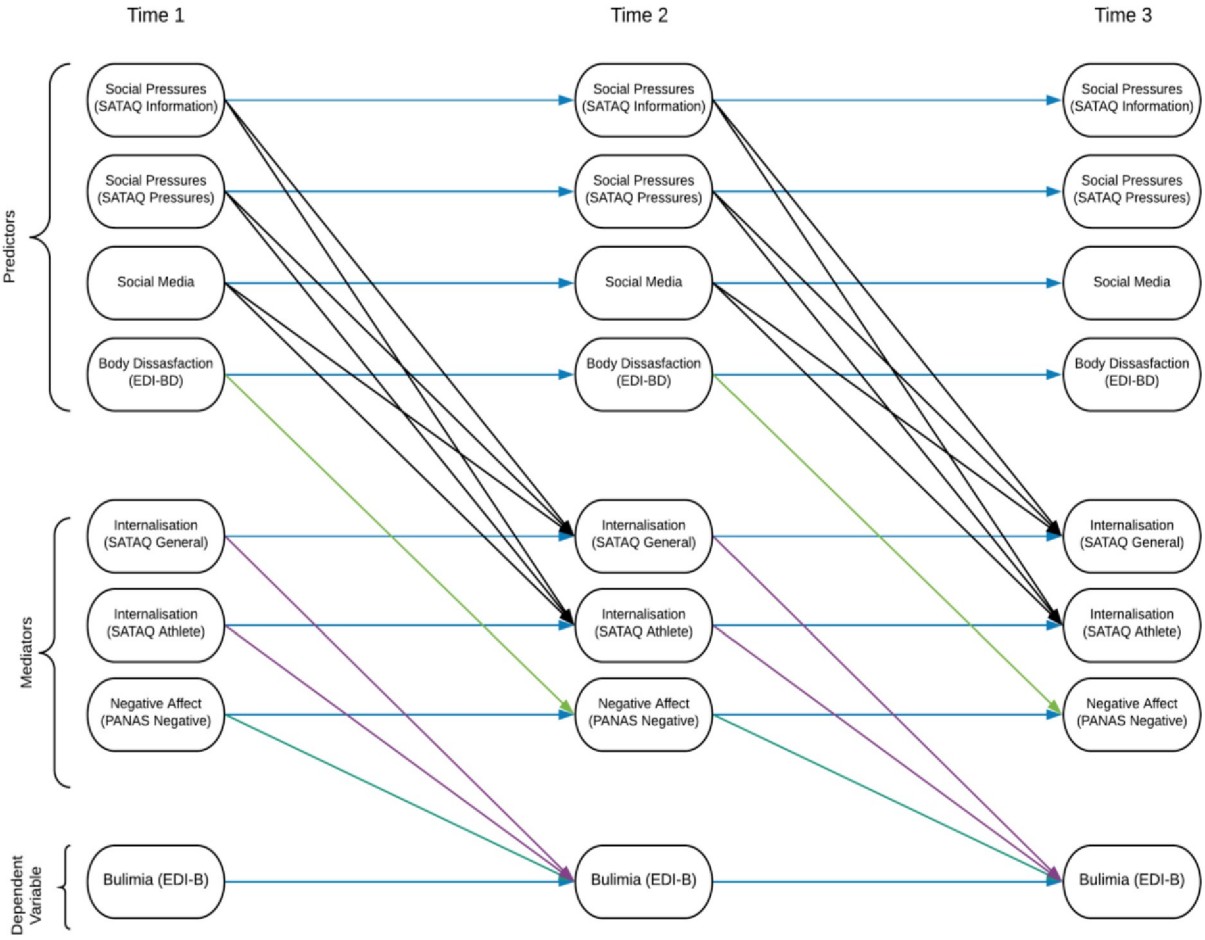

**Fig 2. Cross-lag mediation model.**

data on the same cohort at two further time points, six months apart, enabling the testing of a cross-lagged type mediation model [32]. The primary hypothesis is that the model devised at T1 will remain of good fit across time, when extended to this cross-lagged form as illustrated in Fig 2 below. The second hypothesis is that each path will constitute a significant effect, with Body Dissatisfaction at time 1 having a positive and significant effect on Bulimia at Time 3, mediated by Negative Affect at Time 2; and Social Pressures at Time 1 having a positive and significant relationship with Time 3 Bulimia mediated by Internalisation at Time 2. Finally, as gender and type of sport play a moderating role in the susceptibility of an athlete to developing disordered eating, it is pertinent to explore these factors [33, 34]. As such it is also hypothesised that the predictor to mediator relationships in the model will be significantly moderated by both gender and lean/nonlean sport type.

## Methods

### Procedure

Data collection took place at three time points over the course of a year. Time point one (T1) as described by Stoyel and colleagues was a three-week period in February 2019, time point two (T2) was a three-week period in October 2019, and time point three (T3) was a three-week period in February 2020 [31]. The study received ethical approval from the Clinical

Educational and Health Psychology Department at University College London, reference for this approval: CEHP/2018/573.

All data was collected online, using Opinio (opinion.ucl.ac.uk) for T1 and Qualtrics (www.uclpsych.eu.qualtrics.com) for T2 and T3. This research engaged a volunteer sample via social media and the author's connections as a sport psychologist at local sporting clubs. The inclusion criteria, set at T1, stated that participants had to be over the age 18; had to consider themselves to be an athlete (determined with a simple yes/no answer to the question "Do you identify as an athlete?"); had to be training for a minimum of ten hours a week; and had to be actively competing. These criteria were set such that those included in the study were athletes, rather than just regular exercisers.

At T1 1017 participants responded (and their data were used in the cross-sectional study described in citation [31]). Participants only completed the questionnaires if they had provided written consent via a consent form which they completed online. T1 participants who had given consent for follow-up contact were invited via email to participate at T2. At T2, 879 responses were collected. Those who responded at T2 were then asked to participate again at T3, at which 744 responses were collected. From T1 to T3 there was a 26.8% attrition rate. At each time point, participants gave informed consent, and received a £5 voucher for completing the questionnaire. All those participants who completed the questionnaire too quickly based on analysis of those in the outer quartiles of a normal distribution curve (under eight minutes at T1 and under six minutes at T2 and T3 as the second and third timepoint questionnaires were shorter) were removed, giving a final analysis sample of 802 observations, of whom 802 responded at T1, 551 at T2 and 469 at T3.

## Participants

This research engaged a volunteer sample via social media and the author's connections as a sport psychologist at local sporting clubs. The inclusion criteria at T1 were that participants had to be over the age of 18; had to consider themselves to be an athlete (determined with a simple yes/no answer to the question "Do you identify as an athlete?"); had to be training for a minimum of ten hours a week; and had to be actively competing. These criteria were set such that those included in the study were athletes, rather than just regular exercisers.

The sample was made up of 54.9% males and 45.1% females. The majority were aged between 18 and 26 (84.9%) with 15.1% of the sample aged 27 years or older at T1. Other sample characteristics are given in Table 1. At T1, main sports were basketball 19.6%, swimming 17.6% distance running 11.2%, football 11.8%, dancing 8.3%, tennis 7.5%, volleyball 8.5%, 7.6% other track and field event, with other sports represented with smaller percentages including hockey, badminton, lacrosse, cricket, cycling, golf, triathlon, rugby, boxing, and rowing. Body Mass Index (BMI) and EDE-Q global scores (used for diagnostic purposes of eating disorders) are also presented in Table 1. The mean EDE-Q global scores and is considered healthy and not indicative of an eating disorder. The range of BMI scores is also presented, with the percentages for those that fall outside the healthy range (below 18.5 and above 25) also included. While, the extreme BMI scores do indicate an unhealthy weight to height ratio for some participants and many fall outside the set indicators for healthy BMI, it is important to note that BMI is an insensitive and inaccurate indicator for those with large muscle mass as is likely the case with a sample of athletes [35].

## Measures

To capture both social pressures and internalisation the third edition of the sociocultural attitudes towards appearance was utilised (SATAQ; [36]). The SATAQ internalisation dimension has both general and athlete subscales and has exhibited good measurement properties in

**Table 1. Descriptive statistics and clinical information.**

| Descriptive Variable | Time 1 | Time 2 | Time 3 |
|---|---|---|---|
| Lean/Nonlean Sport | 59.1% nonlean | 44.0% nonlean | 48.3% nonlean |
| | 40.1% lean | 56.0% lean | 51.5% lean |
| Hours/Week | 10–15 hrs: 20.3% | 10–15 hrs: 31.6% | 10–15 hrs: 42.6% |
| | 16–25 hrs: 58.1% | 16–25 hrs: 67.5% | 16–25 hrs: 56.3% |
| | 26+ hrs: 21.6% | 26+ hrs: 0.9% | 26+ hrs: 1.1% |
| Level | Nonelite: 83.5% | Nonelite: 53.3% | Nonelite: 39.0% |
| | Elite: 16.5% | Elite: 46.6% | Elite: 61.0% |
| Years done sport | 1–3 years: 8.6% | N/A | 1–3 years: 12.2% |
| | 4–8 years: 58.1% | | 4–8 years: 62.4% |
| | 9–15 years: 32.3% | | 9–15 years: 24.9% |
| | 16+years: 1% | | 16+years: 0.4% |
| In season/off season | 77.2% in season | 57.1% in season | 66.1% in season |
| | 22.8% off season | 42.9 off season | 33.9% off season |
| EDE-Q Global | M = 2.26 (SD = .85) | M = 2.00 (.89) | M = 1.59 (SD = .91) |
| BMI | Range: 14.15–35.22 | Range: 15.19–36.35 | Range: 14.36–36.79 |
| | M = 22.41 (SD = 3.46) 13.5% below 18.5; 20% over 25 | M = 22.64 (SD = 4.16) | M = 23.65 (SD = 4.97) |
| | | 11.4% below 18.5; 23.6% above 25 | 14.7% below 18.5; 38.2 above 25 |

\* Nonelite = regional & county; Elite = national & international; T2 & T3 'valid percent' was used.

previous studies [37]. Likewise, the SATAQ subscales of Pressures and Information [along with the social media questions] were used to measure the risk factor Social Pressures in Petrie and Greenleaf's model, alongside additional questions that the research team developed in an attempt to modernise the scale by capturing social media pressures. These new items matched the typical wording of the SATAQ [36]. For example, "Social media is an important source of information about fashion and 'being attractive.'" Specifically, the SATAQ subscales of Pressures and Information (along with the social media questions) were used to measure the risk factor Societal Pressures in Petrie and Greenleaf's model with both the General and Athlete Internalisation subscales of the SATAQ used to measure Internalisation. A full listing of items is given in Table 3. Response coding for these items were Definitely Disagree, Mostly Disagree, Neither Agree nor Disagree, Mostly Agree and Definitely Agree.

The ten negative items in the Positive and Negative Affect Schedule (PANAS) were used to measure Negative Affect [38, 39]. A full listing of items is given in Table 3. Response coding asked about emotions and feelings and the extent to which they were felt was from Very slightly or not at all (1) to Extremely (5) over the past week.

To capture the experiences of body dissatisfaction and bulimia related symptomatology, the 9-item Eating Disorder Inventory Body Dissatisfaction (EDI-BD) and the 7-item Bulimia (EDI-B) subscales were utilised respectively [40, 41]. A full listing of items is given in Table 3. Response coding for both the subscales was Never, Rarely, Sometimes, Often, Usually, Always. Scoring was such that numerical scores of 000123 were applied respectively (and the reverse for reversed scored).

The across-time reliabilities of all scales can be found in Table 2 and the items for each scale can be found in Table 3.

## Data preparation

The data was cleaned and prepared in IBM SPSS (version 25) before formal analysis. As the first step in data preparation, across-time reliabilities were calculated (see Table 2). The

Table 2. Across time reliabilities.

| Factor | Items | Reverse items | Adjustment suggested | Pre-adjustment of scale | | | Post-adjustment of scale | | | Conclusion |
|---|---|---|---|---|---|---|---|---|---|---|
| | | | | alpha t1 | alpha t2 | alpha t3 | alpha t1 | alpha t2 | alpha t3 | |
| Body Dissatisfaction | 1–9 | 3,4,5,7,9 | Drop rev items | 0.29 | 0.26 | 0.23 | 0.6 | 0.52 | 0.58 | Use 1 2 6 8 |
| Bulimia | 1–7 | -- | NA | 0.75 | 0.59 | 0.7 | -- | -- | -- | Use all |
| Negative Affect | 1–10 | | NA | 0.89 | 0.68 | 0.76 | -- | -- | -- | Use all |
| Social Pressures (Information) | 1–9 | 3, 4, 8 | Drop rev items | 0.3 | 0.48 | 0.22 | 0.77 | 0.65 | 0.41 | Use items 1,2,5,6,7,9 |
| Social Pressures (Pressures) | 1–7 | 2 | Drop rev item | 0.59 | 0.56 | 0.4 | 0.77 | 0.63 | 0.47 | Use items 1, 3–7 |
| Internalisation General | 1–9 | 1, 6, 9 | Drop rev items | 0.37 | 0.53 | 0.33 | 0.76 | 0.63 | 0.46 | Use items 2–5 7, 8 |
| Internalisation Athlete | 1–5 | 1 | Drop rev item | 0.41 | 0.5 | 0.25 | 0.7 | 0.56 | 0.34 | Use items 2–5 |
| Social Media | 5–9 | 5 | -- | 0.70 | 0.33 | 0.52 | -- | -- | -- | Use all |

reliabilities for all the scales except for Bulimia (EDI-B) and Negative Affect (PANAS Negative Items) were below an acceptable standard for use in analysis. Therefore, two steps were taken. Firstly, all those participants who completed the questionnaire too quickly based on analysis of those in the outer quartiles of a normal distribution curve were removed giving a final analysis sample of 802 observations, of whom 802 responded at T1, 551 at T2 and 469 at T3. 'Too quickly' was defined as under eight minutes at T1 and under six minutes at T2 and T3 (as the second and third timepoint questionnaires were shorter).

Secondly, reverse-scored items were eliminated. As seen in Table 2, this removal did improve the reliabilities of the scales to be utilised in analysis. Details of each scale, those items completed by participants and then subsequently used in analysis can be found in the item map (see Table 3).

## Data analysis

Firstly, confirmatory factor analyses (CFA) to examine the structural validity, the temporal (across time) invariance, and the multigroup invariance (between genders and lean/non-lean sports) of the proposed measurement model and the scales within it. Given the multiple time points and large number of items within each scale, a global CFA of all scales at once would have resulted in an unsatisfactorily low case-free parameter ratio, hence the Social Pressure, Negative Affect, Bulimia, Body Dissatisfaction and Internalisation measures were each considered separately. Having made any necessary adjustments, reliability analyses were then conducted (Cronbach's alpha coefficient) to check the internal consistency of the proposed scales.

Table 3.

| Model Component | Scale Used | Items Asked: | Items Included in Analysis |
|---|---|---|---|
| Body Dissatisfaction | EDI-BD | I think that my stomach is too big | I think that my stomach is too big |
| | | I think that my thighs are too large | I think that my thighs are too large |
| | | **I think that my stomach is just the right size** | I think my hips are too big |
| | | **I feel satisfied with the shape of my body** | I think my buttocks are too large |
| | | **I like the shape of my buttocks** | |
| | | I think my hips are too big | |
| | | **I think that my thighs are just the right size** | |
| | | I think my buttocks are too large | |
| | | **I think that my hips are just the right size** | |

*(Continued)*

**Table 3.** (Continued)

| Model Component | Scale Used | Items Asked: | Items Included in Analysis |
|---|---|---|---|
| Negative Affect | PANAS Negative | Distressed | Distressed |
| | | Upset | Upset |
| | | Guilty | Guilty |
| | | Scared | Scared |
| | | Hostile | Hostile |
| | | Irritable | Irritable |
| | | Ashamed | Ashamed |
| | | Nervous | Nervous |
| | | Jittery | Jittery |
| | | Afraid | Afraid |
| Societal Pressures | SATAQ Pressures | I've felt pressure from TV or magazines to lose weight | I've felt pressure from TV or magazines to have a perfect body |
| | | I've felt pressure from TV and magazines to be thin | I've felt pressure from TV or magazines to diet |
| | | I've felt pressure from TV or magazines to have a perfect body | I've felt pressure from TV or magazines to exercise |
| | | I've felt pressure from TV or magazines to diet | I've felt pressure from TV or magazines to change my appearance |
| | | I've felt pressure from TV or magazines to exercise | |
| | | I've felt pressure from TV or magazines to change my appearance | |
| | | TV programmes are an important source of information about fashion and "being attractive" | TV programmes are an important source of information about fashion and "being attractive" |
| | SATAQ Information | TV commercials are an important source of information about fashion and "being attractive" | TV commercials are an important source of information about fashion and "being attractive" |
| | | **Music videos on TV are not an important source of information about fashion and "being attractive"** | Magazine advertisements are an important source of information about fashion and "being attractive" |
| | | **Magazine articles are not an important source of information about fashion and "being attractive"** | Pictures in magazines are an important source of information about fashion and "being attractive" |
| | | Magazine advertisements are an important source of information about fashion and "being attractive" | Movies are an important source of information about fashion and "being attractive" <br> Famous people are an important source of information about fashion and "being attractive" |
| | | Pictures in magazines are an important source of information about fashion and "being attractive" | |
| | | Movies are an important source of information about fashion and "being attractive" | |
| | | **Movie stars are not an important source of information about fashion and "being attractive"** | |
| | | Famous people are an important source of information about fashion and "being attractive" | |
| | | Social media is an important source of information about fashion and "being attractive | |
| | Social Media | I compare my appearance to the appearance people on social media | Social media is an important source of information about fashion and "being attractive |
| | | I've felt pressure from social media to be thin | I've felt pressure from social media to be thin |
| | | I wish I looked like the influencers on social media | I wish I looked like the influencers on social media |
| | | I compare my life to the life portrayed by people on social media | |

(*Continued*)

**Table 3.** (Continued)

| Model Component | Scale Used | Items Asked: | Items Included in Analysis |
|---|---|---|---|
| Internalisation | SATAQ internalisation General; | **I do not care if my body looks like the body of people who are on TV** | I compare my body to the bodies of people who are on TV |
| | | I compare my body to the bodies of people who are on TV | I would like my body to look like the models who appear in magazines |
| | | I would like my body to look like the models who appear in magazines | I compare my appearance to the appearance of TV and movie stars |
| | | I compare my appearance to the appearance of TV and movie stars | I would like my body to look like the people who are in movies |
| | | I would like my body to look like the people who are in movies | I wish I looked like the models in music videos |
| | | **I do not compare my body to the bodies of people who appear in magazines** | I compare my appearance to the appearance of people in magazines |
| | | I wish I looked like the models in music videos | |
| | | I compare my appearance to the appearance of people in magazines | |
| | | **I do not try to look like the people on TV** | |
| | SATAQ internalisation athlete | **I do not wish to look as athletic as the people in magazines** | |
| | | I compare my body to that of people in "good shape" | I compare my body to that of people in "good shape" |
| | | I wish I looked as athletic as sports stars | I wish I looked as athletic as sports stars |
| | | I compare my body to that of people who are athletic | I compare my body to that of people who are athletic |
| | | I try to look like sports athletes | I try to look like sports athletes |
| Bulimia | EDI-B | I eat when I am upset | I eat when I am upset |
| | | I stuff myself with food | I have gone on eating binges where I have felt that I could not stop |
| | | I have gone on eating binges where I have felt that I could not stop | I think about bingeing (overeating) |
| | | I think about bingeing (overeating) | I eat moderately in front of others and stuff myself when they're gone |
| | | I eat moderately in front of others and stuff myself when they're gone | I have the thought of trying to vomit in order to lose weight |
| | | I have the thought of trying to vomit in order to lose weight | I eat or drink in secrecy |
| | | I eat or drink in secrecy | |

Bolded indicates reversed items.

Secondly, having calculated mean scale (i.e. composite) scores for each construct measured, a series of path analysis models was then fitted to test the hypothesised cross-lagged mediation model, and its moderation by gender and lean/nonlean sport (Fig 2). When fitting the cross-lagged mediation model, analysis began with a model in which paths were free to differ across time. The stability of paths across time (stationarity) was then tested by fixing equal across time, in sequence: T1 to T2 and T2 to T3 autoregressive paths between the same variables across time; T1 to T2 and T2 to T3 paths from the predictors (social pressures and body dissatisfaction) to the mediators (Internalisation and Negative Affect); and T1 to T2 and T2 to T3 paths from the mediators to the outcome (Bulimia). The aim was to show that these fixings did not significantly depreciate model fit, i.e. there was no evidence of variation in these relationships across time. The indirect effects from societal pressures and body dissatisfaction to the outcome of bulimia were then calculated to test whether mediation via internalisation and negative affect respectively was occurring as hypothesised [42].

Finally, the moderation hypotheses were tested by adding the interaction effects of both gender and lean/non-lean sport with societal pressures on mediator internalisation, and with body dissatisfaction on mediator negative affect.

Mplus was used to fit and test between models, with Full Information Maximum likelihood estimation employed on all cases [43]. A *p level of* < 0.005 used throughout.

Item Map (Table 3)

## Results

Table 4 gives the results of the CFA for each of the five constructs of the model, showing both the fit of proposed number of factors with no restrictions in loadings and intercept parameters across time (configural temporal invariance), and the comparisons with models in which equivalent loadings and then intercepts were fixed equal across time (metric temporal and scalar temporal invariance). To achieve a satisfactory fit for the configural invariance model, a small number of items were dropped (detailed in Table 3 above), almost all negatively worded items. Table 5 gives the equivalent tests when considering the multigroup invariance of these respective measurement models between genders and between participants of lean and non-lean sports.

For each of Negative Affect, Body Dissatisfaction, and Bulimia the proposed 1 factor models provided an adequate fit to the data, which was not significantly compromised by fixing factor loadings equal across time. This suggest that the understanding of the items as measures of the respective concepts was consistent across time. Comparing the configural and metric invariance for Negative Affect shows a chi-square difference from 571.00 to 594.18 and no change in the degrees of freedom which was not a significant change with $p = 0.18$. Scalar temporal invariance, how item responses were understood across time, was not achieved for Bulimia and Body Dissatisfaction–however the fit of the Body Dissatisfaction with scalar invariance, though significantly worse than the metric invariance model, was still satisfactory with the CFI greater than 0.90 for metric and configural models (see Table 4).

For internalisation, a two-factor model of general and athlete dimensions offered a satisfactory fit with a CFI of 0.91 for both configural models and outperformed a potentially competing factor model. When applying metric invariance, the fit of the two-factor model was not significantly reduced with the CFI remaining at 0.91, but scalar invariance was not achieved.

**Table 4. Confirmatory factor analyses.**

| Scale | Model | chi-sq | df | chi-sq difference | ch df | p | CFI | RMSEA | SRMR | chisq—df ratio |
|---|---|---|---|---|---|---|---|---|---|---|
| Negative Affect | Configural | 571.00 | 402 | -- | -- | -- | 0.89 | 0.03 | 0.05 | 1.42 |
| | Metric | 594.18 | 420 | 23.19 | 18 | 0.18 | 0.89 | 0.03 | 0.06 | 1.41 |
| | Scalar | 702.00 | 438 | 107.82 | 18 | <0.01 | 0.83 | 0.04 | 0.06 | 1.60 |
| Bulimia | Configural | 140.31 | 132 | -- | -- | -- | 0.99 | 0.01 | 0.04 | 1.06 |
| | Metric | 163.79 | 142 | 23.49 | 10 | 0.01 | 0.97 | 0.02 | 0.04 | 1.15 |
| | Scalar | 183.95 | 152 | 20.16 | 10 | 0.03 | 0.96 | 0.02 | 0.05 | 1.21 |
| Body dissatisfaction | Configural | 62.42 | 51 | -- | -- | -- | 0.97 | 0.02 | 0.04 | 1.22 |
| | Metric | 80.88 | 57 | 18.46 | 6 | 0.01 | 0.93 | 0.03 | 0.04 | 1.41 |
| | Scalar | 120.10 | 63 | 39.21 | 6 | <0.01 | 0.83 | 0.05 | 0.05 | 1.91 |
| Internalisation | Configural | 522.70 | 390 | -- | -- | -- | 0.91 | 0.03 | 0.02 | 1.34 |
| (2 factors) | Metric | 546.34 | 406 | 23.64 | 16 | 0.10 | 0.91 | 0.03 | 0.05 | 1.35 |
| | Scalar | 582.73 | 422 | 36.39 | 16 | <0.01 | 0.89 | 0.03 | 0.05 | 1.38 |
| Societal Pressure | Configural | 976.93 | 783 | -- | -- | -- | 0.91 | 0.02 | 0.04 | 1.25 |
| (3 factors) | Metric | 1004.92 | 805 | 27.98 | 22 | 0.18 | 0.91 | 0.02 | 0.05 | 1.25 |
| | Scalar | 1066.35 | 827 | 61.43 | 22 | <0.01 | 0.89 | 0.03 | 0.05 | 1.29 |

**Table 5. Group Invariance for gender and lean/nonlean.**

| Scale | Model | chi-sq | df | ch chi-sq | ch df | p | CFI | RMSEA | SRMR | chisq—df ratio |
|---|---|---|---|---|---|---|---|---|---|---|
| Negative Affect | *Lean vs Nonlean* | | | | | | | | | |
| | Configural | 1068.74 | 840.00 | | | | 0.86 | 0.04 | 0.07 | 1.27 |
| | Metric | 1075.32 | 849.00 | 6.58 | 9.00 | 0.68 | 0.86 | 0.04 | 0.07 | 1.27 |
| | Scalar | 1109.06 | 876.00 | 33.74 | 27.00 | 0.17 | 0.85 | 0.04 | 0.07 | 1.27 |
| | *Gender* | | | | | | | | | |
| | Configural | 1082.57 | 840.00 | | | | 0.85 | 0.04 | 0.07 | 1.29 |
| | Metric | 1095.16 | 849.00 | 12.59 | 9.00 | 0.18 | 0.85 | 0.04 | 0.07 | 1.29 |
| | Scalar | 1123.76 | 876.00 | 28.61 | 27.00 | 0.38 | 0.85 | 0.04 | 0.07 | 1.28 |
| Internalisation | *Lean vs Nonlean* | | | | | | | | | |
| | Configural | 986.81 | 812.00 | | | | 0.89 | 0.03 | 0.06 | 1.22 |
| | Metric | 998.69 | 820.00 | 11.88 | 8.00 | 0.16 | 0.88 | 0.03 | 0.06 | 1.22 |
| | Scalar | 1011.83 | 844.00 | 13.15 | 24.00 | 0.96 | 0.89 | 0.03 | 0.06 | 1.20 |
| | *Gender* | | | | | | | | | |
| | Configural | 945.19 | 812.00 | | | | 0.91 | 0.03 | 0.06 | 1.16 |
| | Metric | 954.79 | 820.00 | 9.60 | 8.00 | 0.29 | 0.91 | 0.03 | 0.06 | 1.16 |
| | Scalar | 976.29 | 844.00 | 21.51 | 24.00 | 0.61 | 0.91 | 0.03 | 0.06 | 1.16 |
| Body Dissatisfaction | *Lean vs Nonlean* | | | | | | | | | |
| | Configural | 137.59 | 114.00 | | | | 0.93 | 0.03 | 0.06 | 1.21 |
| | Metric | 144.76 | 117.00 | 7.17 | 3.00 | 0.07 | 0.92 | 0.03 | 0.06 | 1.24 |
| | Scalar | 155.45 | 126.00 | 10.69 | 9.00 | 0.30 | 0.91 | 0.03 | 0.06 | 1.23 |
| | *Gender* | | | | | | | | | |
| | Configural | 143.29 | 114.00 | | | | 0.91 | 0.03 | 0.06 | 1.26 |
| | Metric | 144.45 | 117.00 | 1.16 | 3.00 | 0.76 | 0.91 | 0.03 | 0.06 | 1.24 |
| | Scalar | 153.78 | 126.00 | 9.34 | 9.00 | 0.41 | 0.91 | 0.03 | 0.06 | 1.22 |
| Societal pressure | *Lean vs Nonlean* | | | | | | | | | |
| | Configural | 1907.35 | 1610.00 | | | | 0.86 | 0.03 | 0.06 | 1.19 |
| | Metric | 1918.87 | 1621.00 | 11.52 | 11.00 | 0.40 | 0.86 | 0.03 | 0.06 | 1.18 |
| | Scalar | 1961.78 | 1654.00 | 42.91 | 33.00 | 0.12 | 0.86 | 0.03 | 0.06 | 1.19 |
| | *Gender* | | | | | | | | | |
| | Configural | 1878.17 | 1610.00 | | | | 0.88 | 0.03 | 0.06 | 1.17 |
| | Metric | 1895.98 | 1621.00 | 17.81 | 11.00 | 0.09 | 0.88 | 0.03 | 0.06 | 1.17 |
| | Scalar | 1922.57 | 1654.00 | 26.58 | 33.00 | 0.78 | 0.88 | 0.03 | 0.06 | 1.16 |
| Bulimia | *Lean vs Nonlean* | | | | | | | | | |
| | Configural | 342.52 | 284.00 | | | | 0.93 | 0.03 | 0.06 | 1.21 |
| | Metric | 347.89 | 289.00 | 5.36 | 5.00 | 0.37 | 0.93 | 0.03 | 0.06 | 1.20 |
| | Scalar | 364.93 | 304.00 | 17.04 | 15.00 | 0.32 | 0.92 | 0.03 | 0.06 | 1.20 |
| | *Gender* | | | | | | | | | |
| | Configural | 334.74 | 284.00 | | | | 0.94 | 0.03 | 0.06 | 1.18 |
| | Metric | 359.36 | 289.00 | 24.62 | 5.00 | 0.00 | 0.91 | 0.03 | 0.07 | 1.24 |
| | Scalar | 374.17 | 304.00 | 14.81 | 15.00 | 0.47 | 0.91 | 0.03 | 0.07 | 1.23 |

Note. taking forward metric temporal invariance.

Similarly, the three-factor model for Social Pressures achieved a good level of fit 0.91, outper-formed a one factor model, and achieved metric invariance (see Table 4).

For each of the above measures, the satisfactory metric invariance models were taken for-ward to run a series of multigroup CFAs in which loadings and then intercepts were allowed to vary between male/female athletes, or between lean/non-lean sport athletes. Results of

**Table 6. Path analysis model comparisons, testing stationarity of hypothesised model.**

|  |  | chi-sq | df | ch chi-sq | ch df | p | CFI | RMSEA | SRMR | chisq—df ratio |
|---|---|---|---|---|---|---|---|---|---|---|
| model 1 | Free model | 1903.38 | 179.00 |  |  |  | 0.47 | 0.11 | 0.19 | 10.63 |
| model 2 | fix autoregressive paths equal | 1925.34 | 187.00 | 21.96 | 8 | 0.01 | 0.47 | 0.11 | 0.19 | 10.30 |
| model 3 | fix pred-med paths equal for all | 1939.93 | 194.00 | 14.58 | 7 | 0.04 | 0.47 | 0.11 | 0.19 | 10.00 |
| model 4 | fix med-dv paths equal for all | 2023.96 | 197.00 | 84.03 | 3 | <0.01 | 0.44 | 0.11 | 0.19 | 10.27 |

invariance tests are given in Table 5. The Social Pressures, Internalisation, Body Dissatisfaction and Negative Affect measurement models all exhibited both metric and scalar invariance between groups for both gender and lean/non-lean sport. Bulimia failed to achieve metric invariance between genders, though the fit of the metric invariance model, whilst significantly weaker than the configural invariance model, was itself still satisfactory.

## Structural equation modelling

With evidence for metric invariance and good fit for most scales, the CFAs were subsequently extended to a structural equation model (SEM) also within MPlus. The next step was to test the hypothesised path model, illustrated in Fig 2.

This baseline model with all paths free, model 1, and all of the subsequent models failed to achieve a good fit (see Table 6).

When adding constraints across time to investigate the stationarity of effects, neither fixing the autoregressive pathways (blue paths, Fig 2) nor the predictor to mediator paths (black and light green paths, Fig 2) resulted in a significantly weaker fit. However, fixing the mediators to dependent variable paths (purple and dark green) equal across time significantly reduced the model fit.

Within the final, most constrained model (model 4), there were several significant pathways despite the model lacking an acceptable fit. The pathway from all types of societal pressures to general internalisation was significant. The pathways from internalisation to bulimia were also significant (see Table 7 and Fig 3).

While the most constrained model, model 4, failed to establish a good fit, it did show that, with all paths set equal over time, there is a significant indirect effect of Societal Pressures (operationalised as SATAQ pressures, SATAQ information, and social media questions) on Bulimia, operating via General Internalisation (see Table 8).

Several direct effects were also significant across time also shown in Table 8 and Fig 3. Furthermore, Table 9 shows what parts of the model account for percent change in Bulimia at T3.

## Moderation analyses

Group invariance for lean vs nonlean and gender were tested by taking forward the metric temporal invariance CFA for each scale to assess whether the questions (via metric invariance)

**Table 7. Significant pathways in model 4.**

| Pathway | Estimate | S.E | p-value |
|---|---|---|---|
| Social information to General Internalisation | 0.10 | 0.04 | 0.00 |
| Social Pressures to General Internalisation | 0.12 | 0.04 | 0.00 |
| Social Media to General Internalisation | 0.11 | 0.03 | 0.00 |
| General Internalisation to Bulimia | 0.06 | 0.02 | 0.01 |
| Athlete Internalisation to Bulimia | 0.04 | 0.02 | 0.05 |

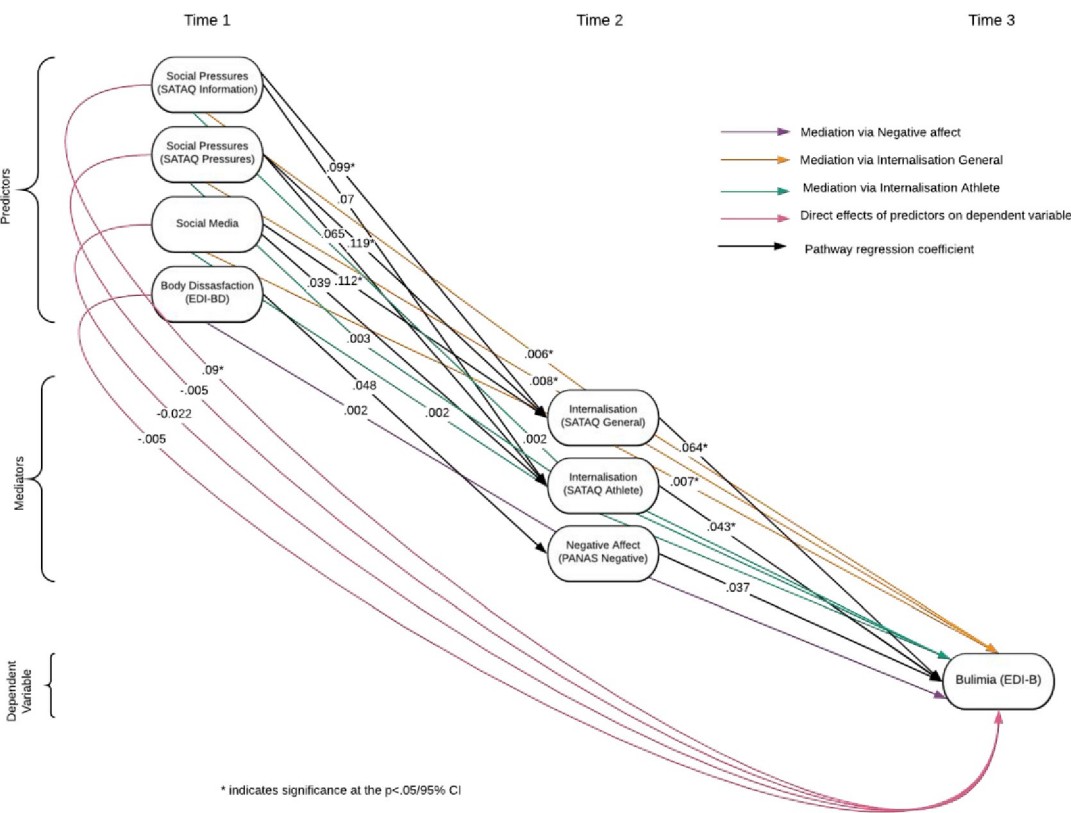

**Fig 3. Visual of indirect, mediated, and direct effects.**

**Table 8. Indirect and direct effects (model 4).**

| | Path | Estimate | S.E. | Est./S.E. | Two-tailed p-value |
|---|---|---|---|---|---|
| Social Media (T1) to Bulimia (T3) Pathway | Direct to General Internalisation (T2) | 0.11 | 0.03 | 3.97 | <0.01* |
| | Direct to Athlete Internalisation (T2) | 0.04 | 0.03 | 1.21 | 0.23 |
| | Via General Internalisation (T2) | 0.01 | 0.00 | 2.24 | 0.03* |
| | Via Athlete Internalisation (T2) | 0.00 | 0.00 | 1.03 | 0.30 |
| | Direct Effect with Bulimia T3 | -0.02 | 0.02 | -0.96 | 0.34 |
| Social Information [T1] to Bulimia [T3] Pathway | Direct to General Internalisation (T2) | 0.10 | 0.04 | 2.86 | 0.00* |
| | Direct to Athlete Internalisation (T2) | 0.05 | 0.04 | 1.20 | 0.23 |
| | Via General Internalisation (T2) | 0.01 | 0.00 | 1.96 | 0.05* |
| | Via Athlete Internalisation (T2) | 0.00 | 0.00 | 1.02 | 0.31 |
| | Direct Effect | 0.09 | 0.03 | 2.90 | 0.00* |
| Social Pressures (T1) to Bulimia (T3) Pathway | Direct to General Internalisation (T2) | 0.12 | 0.04 | 3.36 | 0.00* |
| | Direct to Athlete Internalisation (T2) | 0.07 | 0.04 | 1.53 | 0.13 |
| | Via General Internalisation (T2) | 0.01 | 0.00 | 2.12 | 0.03* |
| | Via Athlete Internalisation (T2) | 0.00 | 0.00 | 1.21 | 0.23 |
| | Direct Effect | -0.01 | 0.03 | -0.14 | 0.89 |
| Body Dissatisfaction (T1) to Bulimia (T3) | Direct to Negative Affect (T2) | 0.05 | 0.03 | 1.64 | 0.10 |
| | Via Negative Affect (T2) | 0.00 | 0.00 | 1.16 | 0.25 |
| | Direct Effect | -0.01 | 0.03 | -0.17 | 0.87 |

*significant at p < .05.

**Table 9.**

| Observed Variable | Estimate ($R^2$) | S.E. | Est./S.E. | Two-tailed p-value |
|---|---|---|---|---|
| Negative Affect T2 | 0.28 | 0.04 | 7.961 | 0.00 |
| Internalisation General T2 | 0.11 | 0.02 | 4.74 | 0.00 |
| Internalisation Athlete T2 | 0.04 | 0.02 | 2.67 | 0.01 |
| Bulimia T2 | 0.13 | 0.03 | 4.74 | 0.00 |
| Negative Affect T3 | 0.06 | 0.01 | 5.24 | 0.00 |
| Internalisation General T3 | 0.05 | 0.01 | 3.26 | 0.00 |
| Internalisation Athlete T3 | 0.01 | 0.01 | 2.39 | 0.02 |
| Bulimia T3 | 0.07 | 0.02 | 3.57 | 0.00 |

and response code (via scalar invariance) are understood the same way across time for all sports/genders (see Table 8). Using chi square of difference testing it was found that all scales had metric and scalar invariance for lean vs nonlean and gender apart from gender invariance for the Bulimia scale.

Finally, to test the final hypothesis and to further extend the model, moderation by gender and lean/nonlean sport was analysed. Moderation of the path from predictors to mediators (black and light green paths in Figs 2 and 3) by gender and lean/nonlean sport showed both gender and lean or nonlean sport consistently moderates the path from Social Pressures (all types) to General Internalisation and from Body Dissatisfaction to Negative Affect. The slope effect of Social Pressures and Body Dissatisfaction is less positive for those in lean sports and for men.

## Discussion

The main aim of this study was to test that T1 model of disordered eating in athletes [31] longitudinally using structural equation modelling in the form of a cross-lagged mediation model.

It was hypothesised that the T1 model would show parsimonious goodness of fit across time in the form of a cross-lag mediation model, but this primary hypothesis failed to be supported, with the model failing to achieve a good overall fit. Specifically, it was hypothesised that Body Dissatisfaction at T1 would have a positive and significant relationship with Bulimia at T3 as mediated by Negative Affect at T2, however, this was not supported. It was also predicted that Societal Pressures at T1 would have a positive, significant relationship with Bulimia at T3, mediated by Internalisation at T2. Part of this hypothesis was supported in that Societal Pressures—in the forms of pressures, information, and social media—significantly predicted bulimic symptomatology a year later, mediated by general internalisation. This suggests that athletes are not only exposed to societal pressures in the form of overt pressure, and information by the society at large, in both mass media and social media, but that it is the internalisation or incorporation of these messages into one's self-worth that predicts bulimic symptomatology. This finding is consistent with results that show that internalisation of the thin ideal, promoted by society and media, in a clinical or nonathlete population is linked to the development of eating disorders [44, 45]. The non-significance of the body dissatisfaction pathway was unexpected. It is possible that the use of general, rather than athlete-specific, body dissatisfaction scales explains this result. Scales that examine general body dissatisfaction may not capture how athletes specifically feel about their body weight and shape. An athlete may feel that their body is satisfactory for societal beauty standards, but not for sport performance and it is possible that the measurement tools in this study did not capture that distinction. Further study of the impact of body dissatisfaction across time is warranted as previous work has been almost exclusively cross sectional (e.g.[23]).

It was hypothesised that the relationships in the model would be significantly moderated by gender and lean/nonlean sport type. This hypothesis was supported with gender and lean/non sport participation both significant moderators, with both gender and lean/nonlean sport participation moderating the pathway from the predictors of societal pressure and body dissatisfaction to the respective T2 mediators of internalisation and negative affect. Therefore, when considering the susceptibility of athletes to developing disordered eating, the type of sport and the gender of that athlete must be taken into account, as the influence of the predictors of social pressure and body dissatisfaction was stronger for female athletes and, surprisingly. for those in nonlean sport. These findings are consistent with previous work that female athletes are more susceptible to the risk factors and correlates of disordered eating as well as disordered eating itself [46, 47]. However, previous work has overwhelmingly, but not always, found that lean sport participants to be more at risk than nonlean (see for example Milligan and Pritchard 2006 versus Kong and Harris 2015 and Torstveit et al 2008). Despite comparisons between lean and nonlean sport, nonlean sport participants are still susceptible to developing disordered eating [21]. The findings in this study perhaps suggest an issue with the classifications of lean and nonlean sports, which was done using the first author's own expertise as an applied Sport Psychologist, but there is no gold-standard way of determining which sports are lean and which are nonlean [22].

## Limitations

Despite its strengths, the current project had several limitations, the first being the poor reliabilities of measurement tools across time. Following the systematic review of Stoyel and colleagues [18], scales with high reliability and frequent usage in the clinical population were chosen in an effort to create consistency. Thus, the first step in exploring why this model did not fit across time was to examine the reliabilities of the scales in this population. Previously, in published clinical, nonathlete samples all scales had reliabilities with $\alpha > 0.8$ [38, 40, 48–50]. However, in the current study, the across-time reliabilities in an athlete sample were poor and worsened across time. Preparation of the data set was needed to increase the reliabilities so that reliable analysis was possible. Interestingly, due to drop out, the make-up of the sample became increasingly elite-heavy across time, perhaps indicating that the scales are less reliable with higher-level competitive athlete samples. Furthermore, confirmatory factor analyses revealed issues with metric and scalar invariance, that while not so ill-fitting that structural equation modelling was invalid, this does show that gold standard measures from non-athlete studies do not perform as well within this athlete sample.

Further limitations include the fact that recruitment was done via social media and participants were rewarded with a voucher. These decisions may have encouraged careless or dishonest responding and meant some items and participants had to be removed from analysis. Additionally, increasing the number of time points either within the year period, or extending the research for a longer period would have allowed for even more sophisticated latent growth curve analysis, as ideally four or more time points are needed for nonlinear analysis [51].

## Implications

This research can inform applied practice, by suggesting that societal pressures and their internalisation could be the focus of prevention interventions for athletes. These findings suggest that mitigating the harmful effects of societal pressures in the form of both information and more overt pressure from an athlete's social sphere and social media is key. While it is not possible to impede all incoming messages by society, a focus for sport psychologists, clinical psychologists, coaches and athletes should be to prevent such messages being internalised into

one's own self-worth. With societal pressure playing a central role in the development of disordered eating, the current study suggests that all types of athletes and both genders are at risk [52–54]. Thus, screening for disordered eating should include all athletes, not just females or those in lean/aesthetic sports. Furthermore, focussing efforts on creating a social environment for all types of athletes that is supportive for wellbeing may be an important aspect of prevention efforts [55, 56]. This finding reiterates that athletes do not exist in a sporting bubble that goes untouched by society's expectations, and furthermore it suggests that interventions developed and tested in nonathlete samples may be able to be adapted for athletes [57]. When working with athletes to mitigate pressures related to disordered eating development we must be aware of those pressures outside of and tangential to the sporting realm.

Future research should focus efforts on developing valid and reliable questionnaires to operationalise disordered eating, eating disorders, and related concepts specifically for athletes. In sum, how we operationalise, measure, and therefore diagnose disordered eating and eating disorders in athletes needs to be tailored to them specifically [58]. It is not necessarily that athletes develop disordered eating symptomatology due to their participation in sport, but the ways we measure and convey disordered eating and related concepts and subsequently intervene may need to be adjusted for athletes. The current scales have items that are not specific for athletes and may therefore under or overestimate disordered eating. For instance, asking an athlete an item such as "Have you had a definite fear that you might gain weight?" as in the EDE-Q, for an athlete who is headed into a lightweight rowing race or wrestling match the following week where there are weight requirements in order to enter the competition, that fear may not be based in disordered eating cognitions but would present as a symptomatic response. In short, there is power in words, and the wording must be appropriate to the context.

## Supporting information

**S1 Data.**
(XLSX)

## Author Contributions

**Formal analysis:** Hannah Stoyel, Chris Stride.

**Supervision:** Lucy Serpell.

**Writing – original draft:** Hannah Stoyel.

**Writing – review & editing:** Chris Stride, Vaithehy Shanmuganathan-Felton, Lucy Serpell.

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
