## [Decision Letter · Decision Letter 0]

8 Apr 2021

PONE-D-21-04001

Understanding risk factors for disordered eating symptomatology

in athletes: a prospective study

PLOS ONE

Dear Dr. Hannah Stoyel,

Thank you for submitting your manuscript to PLOS ONE. After careful consideration, we feel that it has merit but does not fully meet PLOS ONE’s publication criteria as it currently stands. Therefore, we invite you to submit a revised version of the manuscript that addresses the points raised during the review process.

We would appreciate receiving your revised manuscript within 60 days. When you are ready to submit your revision, log on to https://pone.editorialmanager.com/ and select the 'Submissions Needing Revision' folder to locate your manuscript file.

To enhance the reproducibility of your results, we recommend that if applicable you deposit your laboratory protocols in protocols.io, where a protocol can be assigned its own identifier (DOI) such that it can be cited independently in the future. For instructions see: http://journals.plos.org/plosone/s/submission-guidelines#loc-laboratory-protocols

We look forward to receiving your revised manuscript.

Kind regards,

Chiara Milanese

Academic Editor

PLOS ONE

Journal Requirements:

"NO-The funders had no role in the study design, data collection and analysis, decision to publish, or preparation of the manuscript. "

4. We noted in your submission details that a portion of your manuscript may have been presented or published elsewhere.

"Yes, figure 1 and figure 2 have been published before and are needed as reference for clarity in the manuscript. They are appropriately cited and accredited to authors"

Please clarify whether this publication was peer-reviewed and formally published. If this work was previously peer-reviewed and published, in the cover letter please provide the reason that this work does not constitute dual publication and should be included in the current manuscript.

Reviewers' comments:

Reviewer's Responses to Questions

**Comments to the Author**

1. Is the manuscript technically sound, and do the data support the conclusions?

Reviewer #1: Yes

Reviewer #2: Partly

2. Has the statistical analysis been performed appropriately and rigorously? 

Reviewer #1: N/A

Reviewer #2: Yes

3. Have the authors made all data underlying the findings in their manuscript fully available?

Reviewer #1: Yes

Reviewer #2: Yes

4. Is the manuscript presented in an intelligible fashion and written in standard English?

Reviewer #1: Yes

Reviewer #2: Yes

5. Review Comments to the Author

Reviewer #1: The study tested a previous published model of disordered eating in athletes longitudinally.

I have only few suggestions:

Introduction: I suggest removing figure 1 and moving figure 2 (and text) in the method section.

Methods:

1) Please clarify better (possible with references) this sentences: “The inclusion criteria at T1 were that participants had to be over the age of 18; had to consider themselves to be an athlete (determined with a simple yes/no answer to the question “Do you identify as an athlete?”); had to be training for a minimum of ten hours a week; and had to be actively competing. These criteria were set such that those included in the study were athletes, rather than just regular exercisers.”

2) I suggest moving volunteers’ characteristics and Table 1 in the results section.

Please add the strobe statement for observational studies in PLOS Medicine

STROBE Statement: Observational Studies: Getting clear about transparency (strobe-statement.org)

Reviewer #2: General comment

• The paper may have a certain implication, however it is really difficult to read. I advise authors to shorten and focus it.

Specific comments

• The abstract should be better presented I including more numerical results

• The Introduction section is too long, and need to be shortened not to exceed 1 ½ page.

• The Result section is dispersive and should be more focused.

• The quality of the tables and figures should be improved

6. PLOS authors have the option to publish the peer review history of their article (what does this mean?). If published, this will include your full peer review and any attached files.

Reviewer #1: No

Reviewer #2: No

---

## [Author Response · Author response to Decision Letter 0]

18 Jul 2021

PONE-D-21-04001

• A rebuttal letter that responds to each point raised by the academic editor and reviewer(s). This letter should be uploaded as separate file and labeled 'Response to Reviewers'.

• A marked-up copy of your manuscript that highlights changes made to the original version. This file should be uploaded as separate file and labeled 'Revised Manuscript with Track Changes'.

• An unmarked version of your revised paper without tracked changes. This file should be uploaded as separate file and labeled 'Manuscript'.

Further detail of the consent process has been provided in the Method section and in the online submission.

"NO-The funders had no role in the study design, data collection and analysis, decision to publish, or preparation of the manuscript. "

a. Please clarify the sources of funding (financial or material support) for your study. List the grants or organizations that supported your study, including funding received from your institution.

No funding was provided for the study from any grants or organizations. The first author was a self-funded PhD student who provided the renumeration to participants from their own funds.

n/a

n/a

d. If you did not receive any funding for this study, please state: “The authors received no specific funding for this work.”

4. We noted in your submission details that a portion of your manuscript may have been presented or published elsewhere.

"Yes, figure 1 and figure 2 have been published before and are needed as reference for clarity in the manuscript. They are appropriately cited and accredited to authors"

Please clarify whether this publication was peer-reviewed and formally published. If this work was previously peer-reviewed and published, in the cover letter please provide the reason that this work does not constitute dual publication and should be included in the current manuscript.

Reviewer 1 suggested that we remove Figure 1 from the manuscript so we have done this. Figure 2 (now figure 1) was previously published in a peer-reviewed journal, but has been included as this model as it was developed by the current research group and is required in order to explain the research questions in the current study. It has also been appropriately cited. 

We have uploaded the de-identified data set

Reviewer #1: 

1. Introduction: I suggest removing figure 1 and moving figure 2 (and text) in the method section.

Thank you for these suggestions. We have removed figure 1 and renumbered the remaining figures. We have carefully considered whether it is possible to move figure 2 (now re-numbered as figure 1) to the Method, however it does not seem possible to do this because the explanation of the role of the T1 model leads to the hypotheses for the current study. As such this needs to be introduced in the introduction section. 

2. Methods: Please clarify better (possible with references) this sentences: “The inclusion criteria at T1 were that participants had to be over the age of 18; had to consider themselves to be an athlete (determined with a simple yes/no answer to the question “Do you identify as an athlete?”); had to be training for a minimum of ten hours a week; and had to be actively competing. These criteria were set such that those included in the study were athletes, rather than just regular exercisers.”

We have revised the text as follows to clarify inclusion criteria:

Those who identified themselves as athletes and performed at least ten hours of training per week and compete in their current sport were invited to participate. These inclusion criteria were used to determine that those completing the questionnaire were athletes rather than zealous exercisers. Self-identification as an athlete was the primary criterion for inclusion with actively competing also acting as an important determinant of inclusion. As a final back-up, a minimum number of ten hours of participation was set based on the expertise of the first author. 

3. I suggest moving volunteers’ characteristics and Table 1 in the results section.

Thank you, this has been done

4. Please add the strobe statement for observational studies in PLOS Medicine

STROBE Statement: Observational Studies: Getting clear about transparency (strobe-statement.org)

We have examined in detail the various STROBE checklists but there do not appear to be any that are appropriate for a prospective study. 

Reviewer #2: 

1. The paper may have a certain implication, however it is really difficult to read. I advise authors to shorten and focus it.

The paper has been extensively revised to improve readability and considerably shortened

2. The abstract should be better presented I including more numerical results

The abstract has been revised and restructured to make it more concise.

3. The Introduction section is too long, and need to be shortened not to exceed 1 ½ page.

The introduction has been reduced to 1 and a half pages as suggested.

4. The Result section is dispersive and should be more focused.

Efforts have been made to clarify and simplify the results section somewhat. However, it has been difficult to cut much material due to the complex nature of the analysis. We do hope the reviewers agree that it is much improved.

5. The quality of the tables and figures should be improved

Many of the tables have been redesigned to improve layout and readability.

---

## [Decision Letter · Decision Letter 1]

31 Aug 2021

PONE-D-21-04001R1

Understanding risk factors for disordered eating symptomatology

in athletes: a prospective study

PLOS ONE

Dear Dr. Hannah Stoyel,

Thank you for submitting your manuscript to PLOS ONE. After careful consideration, we feel that it has merit but does not fully meet PLOS ONE’s publication criteria as it currently stands. Therefore, we invite you to submit a revised version of the manuscript that addresses the points raised during the review process.

 We would appreciate receiving your revised manuscript within 2 weeks. When you are ready to submit your revision, log on to https://pone.editorialmanager.com/ and select the 'Submissions Needing Revision' folder to locate your manuscript file.

We look forward to receiving your revised manuscript.

Kind regards,

Chiara Milanese

Academic Editor

PLOS ONE

Journal Requirements:

Reviewers' comments:

Reviewer's Responses to Questions

**Comments to the Author**

1. If the authors have adequately addressed your comments raised in a previous round of review and you feel that this manuscript is now acceptable for publication, you may indicate that here to bypass the “Comments to the Author” section, enter your conflict of interest statement in the “Confidential to Editor” section, and submit your "Accept" recommendation.

Reviewer #1: All comments have been addressed

Reviewer #2: All comments have been addressed

2. Is the manuscript technically sound, and do the data support the conclusions?

Reviewer #1: Yes

Reviewer #2: Yes

3. Has the statistical analysis been performed appropriately and rigorously? 

Reviewer #1: N/A

Reviewer #2: Yes

4. Have the authors made all data underlying the findings in their manuscript fully available?

Reviewer #1: (No Response)

Reviewer #2: Yes

5. Is the manuscript presented in an intelligible fashion and written in standard English?

Reviewer #1: (No Response)

Reviewer #2: Yes

6. Review Comments to the Author

Reviewer #1: the marked-up copy of your manuscript that highlights changes made to the original version is correct 'Revised Manuscript with Track Changes', but I feel that the unmarked version of your revised paper without tracked changes has not been uploaded as separate file and labeled 'Manuscript', removing the previous one.

Reviewer #2: (No Response)

7. PLOS authors have the option to publish the peer review history of their article (what does this mean?). If published, this will include your full peer review and any attached files.

Reviewer #1: No

Reviewer #2: No

---

## [Author Response · Author response to Decision Letter 1]

1 Sep 2021

Journal Requirements:

Thank you, the references have been reviewed and no retracted papers have been cited as far as the authors are aware. 

Reviewers' comments:

Reviewer's Responses to Questions

Comments to the Author

1. If the authors have adequately addressed your comments raised in a previous round of review and you feel that this manuscript is now acceptable for publication, you may indicate that here to bypass the “Comments to the Author” section, enter your conflict of interest statement in the “Confidential to Editor” section, and submit your "Accept" recommendation.

Reviewer #1: All comments have been addressed

Reviewer #2: All comments have been addressed

2. Is the manuscript technically sound, and do the data support the conclusions?

Reviewer #1: Yes

Reviewer #2: Yes

3. Has the statistical analysis been performed appropriately and rigorously?

Reviewer #1: N/A

Reviewer #2: Yes

4. Have the authors made all data underlying the findings in their manuscript fully available?

Reviewer #1: (No Response)

Reviewer #2: Yes

5. Is the manuscript presented in an intelligible fashion and written in standard English?

Reviewer #1: (No Response)

Reviewer #2: Yes

6. Review Comments to the Author

Reviewer #1: the marked-up copy of your manuscript that highlights changes made to the original version is correct 'Revised Manuscript with Track Changes', but I feel that the unmarked version of your revised paper without tracked changes has not been uploaded as separate file and labelled 'Manuscript', removing the previous one.

Thank you, this how now been addressed and uploaded correctly. 

Reviewer #2: (No Response)

7. PLOS authors have the option to publish the peer review history of their article (what does this mean?). If published, this will include your full peer review and any attached files.

Do you want your identity to be public for this peer review? For information about this choice, including consent withdrawal, please see our Privacy Policy.

Reviewer #1: No

Reviewer #2: No

---

## [Editor Report · Decision Letter 2]

7 Sep 2021

Understanding risk factors for disordered eating symptomatology

in athletes: a prospective study

PONE-D-21-04001R2

Dear Dr. Hannah Stoyel,

We’re pleased to inform you that your manuscript has been judged scientifically suitable for publication and will be formally accepted for publication once it meets all outstanding technical requirements.

Kind regards,

Chiara Milanese

Academic Editor

PLOS ONE
---

## [Editor Report · Acceptance letter]

14 Sep 2021

PONE-D-21-04001R2 

Understanding risk factors for disordered eating symptomatology in athletes: a prospective study

Dear Dr. Stoyel:

I'm pleased to inform you that your manuscript has been deemed suitable for publication in PLOS ONE. Congratulations! Your manuscript is now with our production department. 

Kind regards, 

on behalf of

Dr. Chiara Milanese 

Academic Editor

PLOS ONE